

# The inconsistent microbiota of *Budu*, the Malaysian fermented anchovy sauce, revealed through 16S amplicon sequencing

Muhammad Zarul Hanifah Md Zoqratt[1] and Han Ming Gan[2,3]

[1] Monash University Malaysia Genomics Facility, School of Science, Jalan Lagoon Selatan, Bandar Sunway, Selangor, Malaysia
[2] GeneSEQ Sdn Bhd, Bukit Beruntung, Selangor, Malaysia
[3] Centre for Integrative Ecology, School of Life and Environmental Sciences, Deakin University, Geelong, Victoria, Australia

## ABSTRACT

*Budu* is a Malaysian fermented anchovy sauce produced by immersing small fishes into a brine solution for 6 to 18 months. Microbial enzymes are known to contribute to fermentation; however, not much is known about the microbial community in *Budu*. Therefore, a better understanding of the *Budu* microbiome is necessary to improve the quality, consistency, and safety of the *Budu* products.

In this study, we collected 60 samples from 20 bottles of *Budu* produced by seven manufacturers. We analyzed their microbiota using V3–V4 16S rRNA amplicon sequencing when we first opened the bottle (month 0), as well as 3 and 7 months post-opening (months 3 and 7). *Tetragenococcus* was the dominant genus in many samples, reaching a maximum proportion of 98.62%, but was found in low abundance, or absent, in other samples. When *Budu* samples were not dominated by a dominant taxa, we observed a wider genera diversity such as *Staphylococcus*, *Acinetobacter*, *Halanaerobium* and *Bacillus*. While the taxonomic composition was relatively stable across sampling periods, samples from two brands showed a sudden increase in relative abundance of the genus *Chromobacterium* at month 7. Based on prediction of metagenome functions, non-*Tetragenococcus*-dominated samples were predicted to have enriched functional pathways related to amino acid metabolism and purine metabolism compared to *Tetragenococcus*-dominated samples; these two pathways are fundamental to fermentation quality and health attributes of fish sauce. Among the non-*Tetragenococcus*-dominated samples, contributions towards amino acid metabolism and purine metabolism were biased towards the dominant taxa when species evenness is low, while in samples with higher species evenness, the contributions towards the two pathways were predicted to be evenly distributed between taxa. Our results demonstrated that the utility of 16S sequencing to assess batch variation in fermented food production. The distinct microbiota was shown to correlate with characteristic metagenome function including functions potentially related to fermented food nutrition and quality.

Corresponding author
Muhammad Zarul Hanifah Md Zoqratt, muhammad.zarulhanifah@monash.edu

## INTRODUCTION

*Budu* is a Malaysian fermented anchovy sauce that is prepared by immersing anchovies in a brine solution (salt concentration 21.5–25.7%) containing natural flavor-enhancing ingredients and left to ferment in earthen containers for 6 to 12 months (*Rosma et al., 2009*). Compared to other Southeast Asian fish sauces like *Nuoc Mam* from Vietnam and *Nampla* from Thailand, which are transparent, *Budu* is turbid and heterogeneous (*Lopetcharat et al., 2001*). Fish sauce is a popular food condiment in Southeast Asia because of its pleasant umami flavor. It is also rich in antioxidants, vitamins and fibrin-clotting inhibitors (*Shivanne Gowda, Narayan & Gopal, 2016*). However, certain metabolites in *Budu* might present as a potential health risk. For instance, high amounts of purine (*Li et al., 2019*) and histamine (*Rosma et al., 2009*) in Budu are associated with gout (*Paul & James, 2017*) and scombroid poisoning (*Tortorella et al., 2014*), respectively.

The fish sauce microbiota can change the fish sauce metabolite content, potentially altering its nutritional value and its flavor. For instance, *Tetragenococcus muriaticus* produces histamine, while *Staphylococcus*, *Bacillus* and *Lactobacillus* can degrade histamine (*Shivanne Gowda, Narayan & Gopal, 2016*). Therefore, a better understanding of the Budu microbial community can elevate the Budu organoleptic and health value by harnessing its microbiota. To the best of our knowledge, all microbiological studies of *Budu* to date were done based on culture-based methods which do not enable an overview of the *Budu* microbial community. For example, *Sim, Chye & Ann (2009)* discovered bacterial succession takes place in *Budu* fermentation from *Micrococcus*-to *Staphylococcus arlettae*-dominated community and uncovered the presence of yeast species such as *Candida glabrata* and *Saccharomyces cerevisiae*. Subsequent microbiological studies on Budu revealed a few other cultivable bacteria such as *Bacillus amyloliquefaciens* FS-05 and *Lactobacillus planatarum* which were able to produce glutamic acid, a compound associated with umami taste (*Zaman et al., 2010*; *Zareian et al., 2012*). A recent study on Malaysian fermented foods also displayed the potential of *Bacillus* sp. in producing biosurfactants that inhibit pathogenic bacterial growth (*Mohd Isa et al., 2020*).

In this study, we surveyed the microbial community of 60 samples from 20 bottles of *Budu* purchased from multiple brands. We investigated their microbial community structure at months 0, 3 and 7 post-opening. We then focused on their predicted metabolic pathways while relating them to microbial composition and diversity.

## MATERIALS & METHODS

### Sample collection

Twenty bottles of *Budu* from different brands were purchased from shops in the state of Kelantan, Malaysia. These 20 bottles were sampled at months 0, 3 and 7 post-opening, amounting to a total of 60 samples (Table 1). The samples were stored at room temperature to emulate typical retail storage conditions. Samples were named according to the following format:<brand><bottle number>_<months since last opened>.

**Table 1 Sample metadata of *Budu* from seven different brands.**

| Brand | Number of biological replicates | Month of purchase | Expiration month | Declared additive(s) | | | | |
|---|---|---|---|---|---|---|---|---|
| | | | | Chili | Tamarind | Lime | Palm Sugar | Sugar |
| A | 3 | July 2016, Nov 2016, Mar 2017 | Feb 2018, May 2018, Sept 2018 | X | | X | X | |
| B | 4 | July 2016, Oct 2016, Nov 2016, Mar 2017 | Feb 2018, Feb 2018, May 2018, Oct 2018 | X | | X | X | |
| C | 4 | July 2016, Oct 2016, Nov 2016, Mar 2017 | Jan 2018, Jan 2018, Jan 2018, Jan 2018 | | X | | | X |
| D** | 3 | July 2016, Nov 2016, Mar 2017 | Jan 2020, Jan 2020, Jan 2020 | | | | | X |
| E | 4 | July 2016, Oct 2016, Nov 2016, Mar 2017 | Jan 2018, Jan 2018, Jan 2018, Jan 2018 | | X | | | X |
| F | 1 | Apr 2017 | Jan 2018 | | X | | | X |
| G | 1 | Apr 2017 | Jan 2018 | | X | | | |

Note:
** Only sample D3 was declared to contain colouring.

## DNA extraction and sequencing

A total of 200 µl of the sample was poured into 1.5 ml Eppendorf tubes and centrifuged at 14,800 rpm for 15 minutes to remove supernatant. The samples were then resuspended in 400 µl of 20 mM Tris-EDTA buffer and added with 100 µm silica beads (OPS Diagnostics LLC, Lebanon, NJ, USA). The samples were later subjected to bead beating using Vortex Genie two Mo Bio (Carlsbad, CA, USA) at 3,200 rpm for 1 h. They were added with 20 µl of 0.5% SDS and 20 µl of 50 mg/ml proteinase K and incubated at 55 °C for 1 h. The samples were added to 100 µl of saturated potassium chloride solution and incubated on ice for 5 minutes. Total DNA was extracted using chloroform-isopropanol precipitation and purified with Agencourt AMPure XP beads (Beckman Coulter, Inc., Indianapolis, IN, USA).

The V3–V4 region of the 16S rRNA gene was amplified using the forward primer 5′-TCGTCGGCAGCGTCAGATGTGTATAAGAGACAGCCTACGGGNGGCWGCAG-3′ and reverse primer 5′ GTCTCGTGGGCTCGGAGATGTGTATAAGAGACAGGACTACHVGGGTATCTAATCC-3′ containing partial Illumina Nextera adapter sequence (*Klindworth et al., 2013*) following the Illumina 16S Metagenomic Sequencing Library Preparation protocol. To enable multiplexing, 16S amplicons were barcoded using different pairs of index barcodes to prepare DNA libraries. DNA libraries were normalized, pooled, denatured and sequenced using Illumina MiSeq (Illumina, San Diego, CA, USA) at Monash University Malaysia Genomics Facility.

## Sequence data analysis, phylogenetic tree construction, taxonomic assignment and generation of feature table

The forward and reverse PCR primer sequences were trimmed off using Cutadapt version 1.16 with default parameters (*Martin, 2011*). Paired-end sequences were merged using fastq_mergepairs, and quality-filtered using fastq_filter (-fastq_maxee: 1.0, -fastq_minlen: 300) in USearch v10.0.240_i86linux32 (*Edgar & Flyvbjerg, 2015*). High-quality sequences

were then denoised to create amplicon sequence variants (ASVs) (File S1). The ASV sequences were used as reference sequences to create a raw feature table, using UNoise3 in USearch (*Edgar, 2016*). Chloroplast and mitochondrial ASVs were filtered out from the feature table and later rarefied to 6,000 reads per sample for downstream analyses (File S3).

Multiple sequence alignment of the ASV sequences was conducted using MAFFT while masking unconserved and highly gapped sites (*Katoh & Standley, 2013*). A phylogenetic tree was then constructed from aligned ASV sequences and was rooted at midpoint using FastTree version 2.2.10 (*Bolyen et al., 2019*; *Price, Dehal & Arkin, 2010*). 16S V3–V4 Naive-Bayes classifier was trained on V3–V4-trimmed 16S sequences of the SILVA 132 release, using q2-classifier plugin in QIIME 2 (*Bokulich et al., 2018*; *Pedregosa et al., 2011*). The SILVA reference sequences were trimmed using the same primer sequences and parameters for the raw sequencing data. The ASV sequences were then taxonomically assigned using the trained classifier (File S2).

## Calculation of alpha and beta diversity indices and comparison of genera distribution between sampling batches

Species richness indices (observed ASVs and Faith PD) as well as species evenness indices (Simpson and Shannon) were calculated using QIIME2 (*Bolyen et al., 2019*; *Hunter, 2007*) at 10 iterations per sequencing depth. Species richness were compared between sampling batches using Wilcoxon paired tests on DABEST-python library (*Ho et al., 2019*). Since there is only one bottle for brands F and G, these samples were not included in alpha diversity statistical comparisons.

Beta diversity was measured based on Bray-Curtis and weighted UniFrac to calculate distances between the microbiota of the samples (*Lozupone et al., 2011*). ASV-and genus-level relative differential abundance between sampling batches were done using QIIME2 plugin ANCOM (*Mandal et al., 2015*).

## Functional and pathway metagenome prediction

Using the normalized feature table as input data, function and pathway prediction pipelines were conducted using PiCrust2 (*Douglas et al., 2020*). Briefly, ASVs were phylogenetically placed into a reference phylogenetic tree using EPA-NG and gappa (*Barbera et al., 2019*; *Czech, Barbera & Stamatakis, 2020*). Hidden state prediction to predict 16S copy number was applied using the castor R package (*Louca & Doebeli, 2018*) to normalize the feature table based on 16S copy number information and to predict the Nearest Sequenced Taxon Index (NSTI) score per ASV. ASVs with NSTI scores higher than 2.0 were assumed to not have a representative genome in the reference phylogenetic tree and thus filtered out from subsequent analyses. Afterwards, prediction of gene family abundance was done against the enzyme commission (EC) database (File S4), followed by prediction of pathway abundance against the METACYC database (File S5) (*Ye & Doak, 2009*). Scores of these predictions were normalized by sequencing depth per sample (6,000 reads).

Predicted pathway table was subjected to Bray Curtis distance normalization, followed by multidimensional scaling (MDS) ordination (ratio transformation) using Ecopy
(https://github.com/Auerilas/ecopy). Samples with high relative abundance of *Tetragenococcus* formed a distinct cluster which separated from non-*Tetragenococcus*-dominated samples. To compare potential pathways that distinguish the two groups, a post-hoc t-test was applied, adjusted by false discovery rate Benjamini Hochberg. A predicted pathway with a corrected *p*-value of below $1 < 10-5$ and Cohen d effect size of at least three is to be considered as significantly different between the *Tetragenococcus*-dominated group and non-*Tetragenococcus*-dominated group (*Terpilowski, 2019*; *Vallat, 2018*).

## Microbial association network

SPIEC-EASI 0.1.4 was used to predict the microbial association network based on co-abundance (*Kurtz et al., 2015*). Since SPIEC-EASI applies its own normalization method, the mitochondria and chloroplast ASVs-filtered unrarefied feature table was used as the input table. ASVs were also filtered by prevalence at the minimum occurrence of 30 samples.

Meinshausen-Bühlmann neighborhood selection model was used to model the microbial interaction network at 10,000 replications (*Meinshausen & Bühlmann, 2006*), and at a variability threshold of 0.05% using StARS (*Liu, Roeder & Wasserman, 2010*). R igraph package was used for network visualization and extracting network properties (*Csardi & Nepusz, 2006*). To find important nodes in the graph, degree centrality and betweenness centrality were two node centrality measures computed from the predicted network. Degree centrality weighs a node's importance by counting the number of edges linked to the node, while betweenness centrality evaluates the geodesic distances from all node pairs that are passing through the particular node. High degree centrality suggests a role as a keystone species, while high betweenness centrality predicts its importance in maintaining the structure of the interaction network.

## RESULTS

### The *Budu* microbiota is inconsistent within the same brand even at high-level taxonomic composition

*Firmicutes*, *Proteobacteria*, *Halanaerobiaeota*, *Actinobacteria*, and *Bacteroidetes* were the top five most abundant phyla across all *Budu* samples (Fig. 1A) with an average relative abundance of 60.26%, 22.86%, 10.60%, 3.91%, and 1.91% respectively, while other lesser phyla contributed less than 0.5%. The cumulative relative abundance of the top five most abundant phyla in each sample ranges from 95.75% to 100.00%. The relative abundance of the dominant phyla was not statistically different between sampling periods (Wilcoxon paired test $p > 0.4$), thus the phyla composition was fairly stable across time.

We observed different trends of phyla composition between brands and within the same brands. For example, all samples from brands B and D were consistently dominated by *Firmicutes*, making up an average relative abundance of 93% in each sample. On the other hand, some samples displayed uneven phyla distribution within a brand. For example, *Halanaerobiaeota* made up a substantial proportion in samples C1, C3

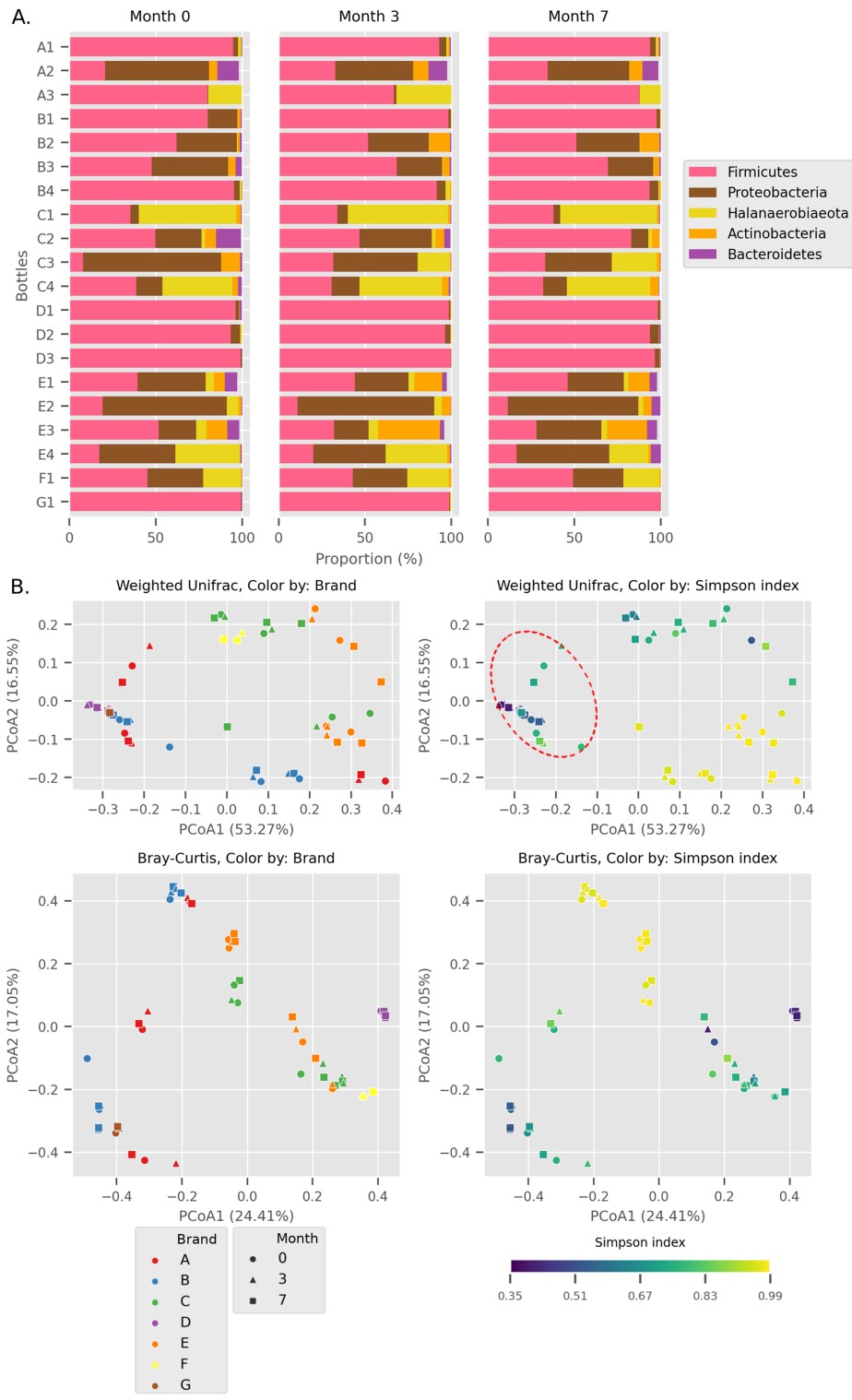

**Figure 1 Overall diversity of the Budu 16S microbiota.** (A) Relative abundance of the top five most abundant phyla in each bottle across sampling time. (B) Principal Coordinates Analysis (PCoA) plots, based on Weighted UniFrac and Bray Curtis distances. Point shapes were assigned by sample batch. Point color were by either brand, or Simpson alpha diversity index. Circular perimeter marks a cluster that was dominated by Firmicutes (>60% relative abundance).

(except sample C3_0) and C4 samples (range 18.8% to 48.4%) but was markedly smaller in C2 samples (average = 2.2%). In another instance, a substantial proportion of *Proteobacteria* and *Halanaerobiaeota* was found in A2 and A3 samples respectively but both phyla were observed in considerably low proportions in A1 samples. The distinct phyla composition of *Budu* microbiota was recapitulated in the Principal Coordinates Analysis (PCoA) plot based on Weighted UniFrac distances (Fig. 1B). For example, there are three clusters of brand A; each cluster consists of three points corresponding to three sampling batches per bottle. This suggests that the microbiota from samples in brand A is similar when they are from the same bottle but is different between bottles. This sub-clustering trend was also observed in brand B, providing further evidence that the *Budu* microbiota is inconsistent in some brands.

In weighted-UniFrac PCoA, samples that were dominated by the *Firmicutes* formed a cluster (Fig. 1B; red dashed outline). This cluster included samples from multiple brands, such as brands D and G. However, in the Bray Curtis distances PCoA, which only weighs relative abundance information and not the phylogenetic relationship between ASVs, this clustering pattern was not observed. Brand D formed an isolated cluster away from other *Firmicutes*-dominated samples in Bray Curtis PCoA. This difference in clustering pattern between weighted-UniFrac and Bray Curtis clustering indicated that brand D samples contained phylogenetically close but distinct ASVs compared to other *Firmicutes*-dominated samples.

In at least half of the samples, 14 genera were found (Fig. 2). Based on the enormous relative abundance (average = 38.77%, max. = 98.62%) and prevalence (96.62%), *Tetragenococcus* was the most dominant genus of the *Budu* microbiota. Although it was prevalent and dominant in some samples, it was also present in low abundances in samples like samples from bottle A2 and samples from brand E. *Halanaerobacterium* was another genus that made up a substantial proportion of some Budu microbiota (max. = 58.15%), despite its lesser prevalence than *Staphylococcus* and *Acinetobacter*. *Tetragenococcus* and *Halanaerobium* were the only two genera that can reach a relative abundance of over 50% of reads per sample–this was observed in 21 samples (brands A, B, D and G) and three samples (brand C), respectively.

Despite the lower overall prevalence of *Weissella* (50%), it had over 20% relative abundance in four samples. This magnitude of relative abundance was not observed in several genera of higher prevalence, such as *Acinetobacter*, *Pseudomonas*, *Psychrobacter*, *Corynebacterium_1*, *Lentibacillus*, *Kocuria*, *Brevibacterium*, and *Paracoccus*. Meanwhile, other prevalent taxa that were present despite their noticeably lower abundance.
For example, the highest relative abundance of *Comamonas* was at most 1.22 % of reads per sample, despite being present in half of the samples. This shows that the prevalence of a genus in the samples is not predictive of its relative abundance. None of the prevalent genera was found in all samples.

Within *Tetragenococcus*, we found two different species: *Tetragenococcus muriaticus* (ASV1) and *Tetragenococcus halophilus* subsp. halophilus (ASV3, ASV4 and ASV7) (Fig. S1). *T. muriaticus* composition was consistently high in brand D, while *T. halophilus*

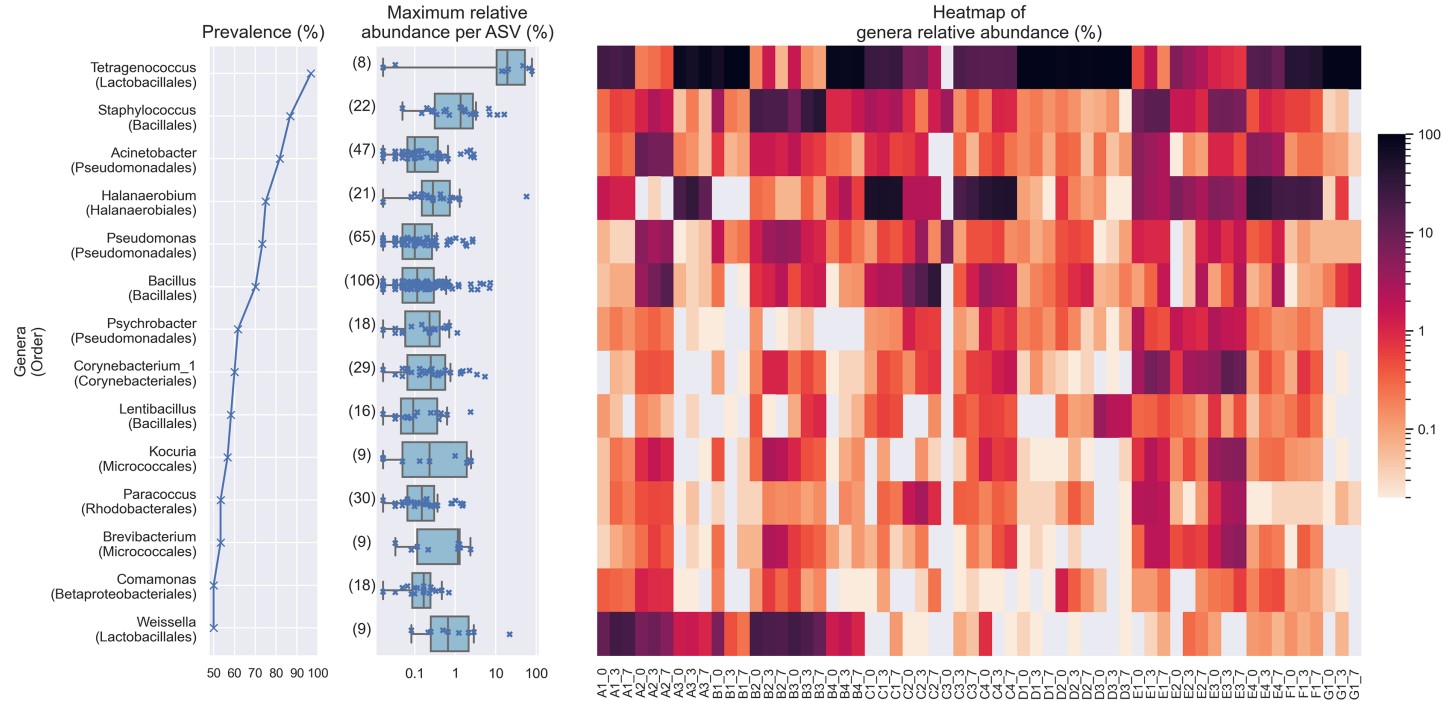

**Figure 2** Prevalence, maximum relative abundance per ASV and relative abundance per samples of prevalent genera (at least 50% prevalence, >0.01% abundance per sample). The bracketed numbers beside the maximum abundance per ASV boxplots indicate the number of ASVs assigned to the genus.

subsp. halophilus was especially abundant in some bottles of brands A, B and G. We also observed the co-presence of both *Tetragenococcus* species in some samples, such as C1 samples and samples from brand F. The consistent, high relative abundance of ASV1 in brand D was likely to explain the sub-cluster formed by brand D samples occurring in the Bray Curtis PCoA (Fig. 1B). *Halanaerobium*, *Staphylococcus* and *Weissella* were the only other genera with ASVs above 10% relative abundance.

*Bacillus* was the most commonly isolated bacterial genus from *Budu* (Mohd Isa et al., 2020; Zaman et al., 2010). The abnormally large number of *Bacillus* ASVs (Fig. 2) could be explained by the historically poor taxonomic demarcation of the *Bacillus* taxonomy (Patel & Gupta, 2020). According to the maximum likelihood phylogenetic tree of partial V3–V4 16S sequences, *Bacillus* ASVs from Budu were represented in different clades within the *Bacillus* genus (Fig. S2). ASV18 was the most abundant and prevalent *Bacillus* ASV, belonging to the Cereus clade (max. = 7.05%). The bootstrap support values of some nodes of the phylogenetic tree were low, ranging from 0.34 to 0.75. Some nodes were positioned into a different clade than expected, such as *Bacillus gibsonii* placed in the Alcalphilus clade. This reflected the limitations of the 16S sequence as a marker to *Bacillus* evolutionary history. The resolution of *Bacillus* taxonomy and functional diversity in *Budu* can be further confirmed in the future using genome sequences of cultivated *Bacillus* isolates or metagenome-assembled genomes from *Budu*.
## Comparison of *Budu* across time reveals minimal differences in alpha diversity, followed by detectable shifts in relative abundance of a few genera

Some *Budu* samples within a brand had considerable different species richness (Figs. 3A and 3B). This was apparent in brands A and E. Such a trend coincides with the inconsistent phyla composition within a brand (Fig. 1). There were also apparent changes in alpha diversity across sampling batches within the bottle. After 3 months, the observed ASVs and Faith PD increased in most bottles of brands A, C and E. Between months 3 and 7, the alpha diversity indices decreased in some of the bottles while others steadily increased. There were also inconsistent shifts in species richness in brand B samples. Fig. S3 shows the rarefaction plots of observed features and the Faith PD indices. Some rarefaction curves did not completely plateau, such as samples from brands D and E, indicating that there may be samples that contain taxa of low relative abundance which were not adequately sampled at sequencing depth of 6,000 reads.

When comparing the genera distribution between sampling batches, *Chromobacterium*, *Barnesiella* and [*Ruminococcus*] *torques* group displayed significantly differential abundance (W statistic > 0.6 × of total features) and F statistic (>10), as seen in Fig. 3C. The relative abundance of *Chromobacterium* was higher at later sampling in samples of brands D and E, reaching above 10% in brand E samples while being virtually absent in other samples (Figs. 3D, S4). *Barnesiella* and *Ruminococcus torques* were relatively more abundant at initial sampling but had a lower relative abundance and did not correlate with any common metadata. Comparison of taxonomic distribution at ASV level between sampling batches resulted in no significantly differential ASV distribution.

## Comparison between Tetragenococcus-dominated and non-Tetragenococcus-dominated microbiota reveals differential enriched predicted pathways, primarily related to amino acid and purine biosynthesis

Based on predicted pathways information, samples dominated by *Tetragenococcus* formed a distinct cluster distant from non-*Tetragenococcus*-dominated samples (PERMANOVA *p*-value 0.001, Fig. 4A). By looking at pathways that were significantly different between the two groups, it was apparent that a few pathways, such as those involved in cell wall biosynthesis pathways and lactose and galactose degradation I (LACTOSECAT-PWY), were enriched in Tetragenococcus-dominated microbiota (Fig. 4B). Meanwhile, various predicted pathways were enriched in non-*Tetragenococcus*-dominated microbiota, especially those involved in purine biosynthesis, amino acid metabolism and vitamin/ cofactor/carrier biosynthesis. Purine and amino acid biosynthesis are implicated with organoleptic quality and health aspects of fermented foods. Therefore, we looked further into non-*Tetragenococcus*-dominated microbiota for predicted contributions of the individual genus to these two pathways.

Aside from *Tetragenococcus*, *Staphylococcus*, *Halanaerobium*, *Bacillus* and *Weissella*, other prevalent genera were not predicted to impact purine and amino acid biosynthesis,

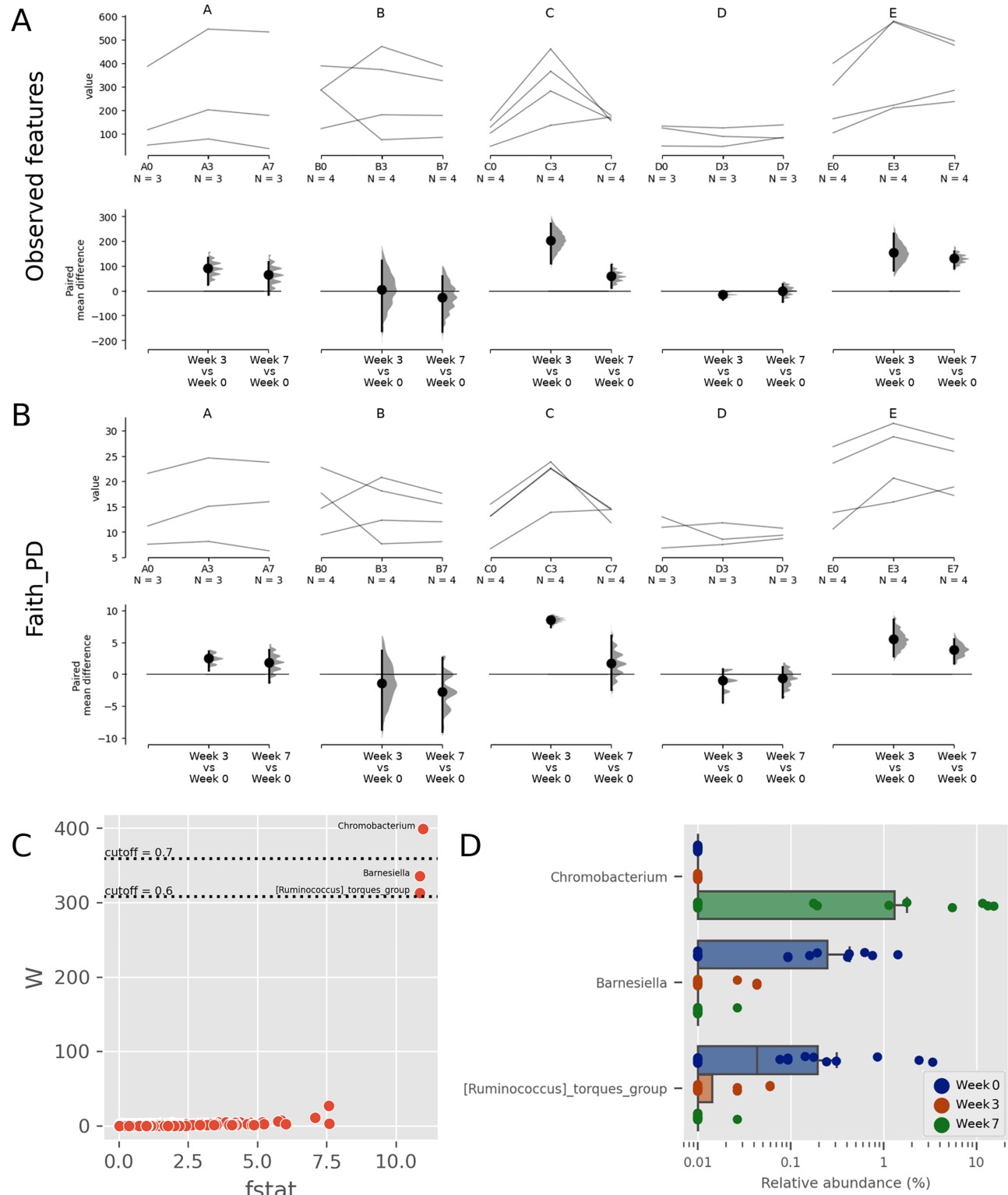

**Figure 3 Trends of alpha diversity indices of the Budu microbiota.** Trajectory of observed ASVs (A) and Faith phylogenetic diversity (B) across sampling periods, and the comparison of alpha diversity measures at $3^{rd}$ and $7^{th}$ month sampling against control range ($0^{th}$ Month), (C) dot plot of W statistic computed by ANCOM against F-stat of each genera, (D) relative abundance of *Chromobacterium*, *Barnesiella* and *[Ruminococcus] torques* group across sampling periods.

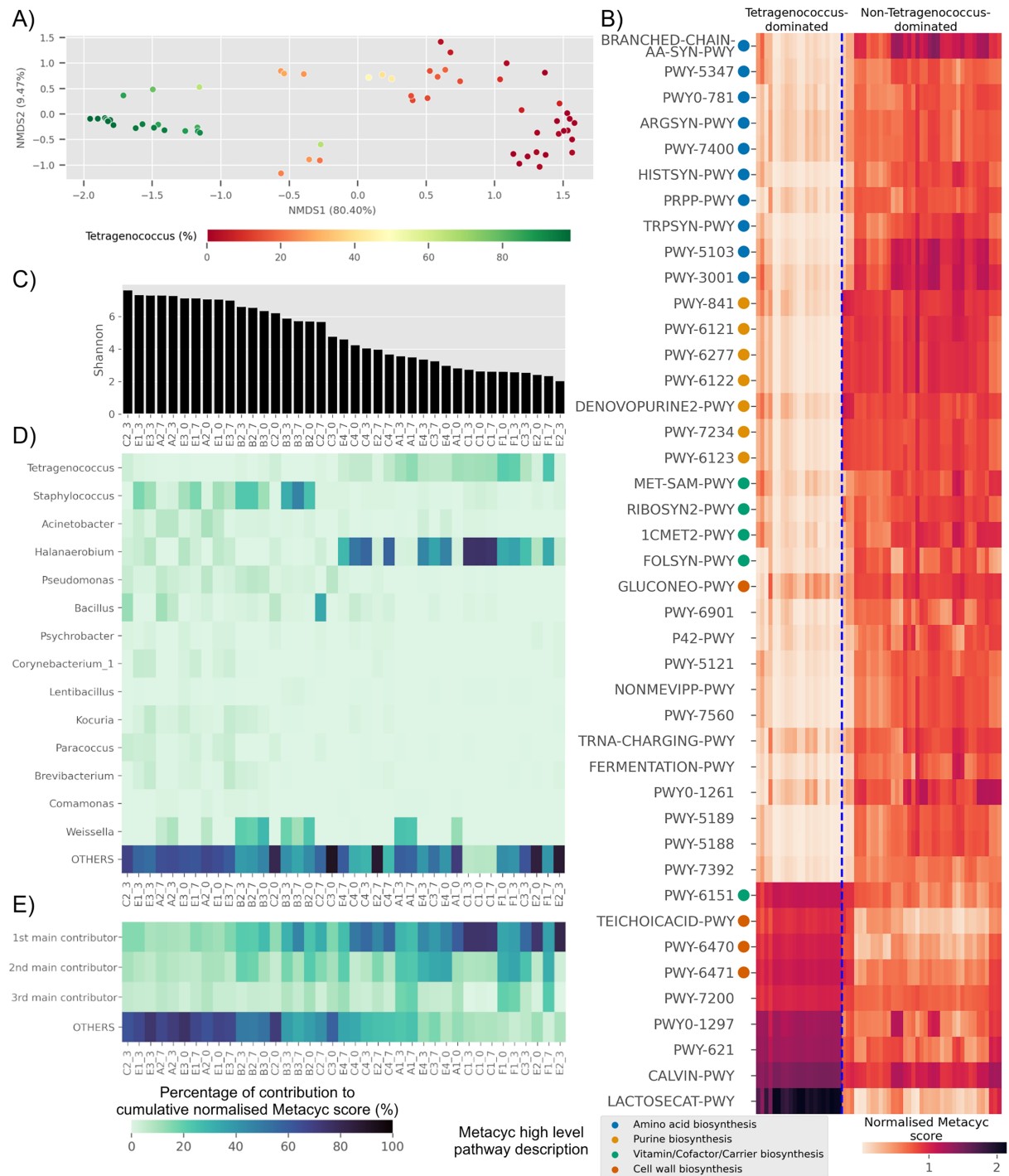

**Figure 4** **Trends of predicted functional diversity correspond to the taxonomic diversity of the underlying Budu microbiota.** (A) Multi-dimensional scaling plot based on overall predicted pathways for each samples. Sample are colored based on relative abundance of *Tetragenococcus*, (B) heatmap of differential pathways between *Tetragenococcus*-dominated and non-*Tetragenococcus*-dominated samples based on normalized METACYC score. Pathways labels are based on shortened pathway labels from the METACYC database, (C) Shannon index, sorted by value. Only non-*Tetragenococcus*-dominated samples are shown, (D) heatmap of pooled contribution towards overall predicted amino acid and purine bio-synthesis pathway per genus of prevalence >= 50% per sample. Only non-*Tetragenococcus*-dominated samples are shown, (E) pooled contribution towards overall predicted amino acid and purine biosynthesis pathway of top three most abundant genus per sample. Only non-*Tetragenococcus*-dominated samples are shown.

relative to the cumulative contribution of the remaining lesser genera (OTHERS) (Fig. 4D). By focusing on the top three most abundant genera per sample regardless of taxonomic prevalence, purine and amino acid biosynthesis in samples of low Shannon index (Fig. 4C) were predicted to be contributed by the dominant genus (Fig. 4E). This included samples without a high prevalence genus, such as samples from bottle E2 (Fig. 4D). This was not observed in samples with high Shannon diversity index. For instance, samples from bottles E1 and E3, which had relatively high Shannon scores, possessed a high percentage of contribution from the remaining genera instead of the top three most abundant genera within these samples (Fig. 4E).

Several predicted pathways were enriched in non-*Tetragenococcus*-dominated samples, such as L-methionine biosynthesis (transsulfuration) (PWY-5347), L-histidine biosynthesis (HISTSYN-PWY), flavin biosynthesis I (bacteria and plants) (RIBOSYN2-PWY) and superpathway of tetrahydrofolate biosynthesis and salvage (FOLSYN-PWY). These pathways are implicated with nutrients that are only obtained through diet.

## Microbial abundance interaction network

After filtering out ASVs present in less than half of the samples, there were a total of 71 nodes in the predicted microbial interaction network (Fig. 5A), 73.7% were positive associations, and 26.3% were negative associations. Several isolated sub-clusters formed, such as a small cluster consisting of four nodes (ASV4, ASV7, ASV10 and ASV11); all four nodes were assigned as *Tetragenococcus halophilus* subsp. Halophilus. The 16S copy number predicted from these ASVs was only one copy per ASV, which suggested that the ASVs belong to closely related *T. halophilus* strains. The isolation of this cluster might imply a lack of interactions between *T. halophilus* subsp. Halophilus and the rest of the *Budu* inhabitants. The nodes were not fully connected with each other, possibly because of the conditional independence implemented in SPIEC-EASI to prevent spurious links. Positive associations were predicted for edges ASV4–ASV7 and ASV7–ASV11. However, a negative association formed between ASV4 and ASV10.

Several nodes of similar taxonomic assignment were predicted to interact with each other, including the earlier mentioned *T. halophilus* subsp. Halophilus sub-cluster as well as ASV1–ASV6 (*Tetragenococcus muriaticus*), ASV47–ASV77–ASV228 (*Acinetobacter*), ASV20–ASV28–ASV53 (*Staphylococcus*), ASV216–ASV133 (*Psychrobacter*) and ASV27–ASV66 (*Kocuria*). Co-abundance between these nodes were expected due to either presence of multiple 16S copies per genome or variation of 16S sequences between microbial strains. There were also patterns of network assortativity, which was most apparent for the Actinobacteria nodes surrounding ASV30 which was assigned as *Brevibacterium*. ASV30 formed a number of positive links with *Corynebacterium* nodes which belonged to the Actinobacteria class. ASV30 formed multiple positive links with non-Actinobacteria nodes belonging to Bacilli and Gammaproteobacteria.

Degree centrality and degree betweenness centrality were two node centrality measures computed from the predicted interaction network. The majority of the nodes had considerably low degrees and possessed low betweenness centrality scores (Fig. 5B).

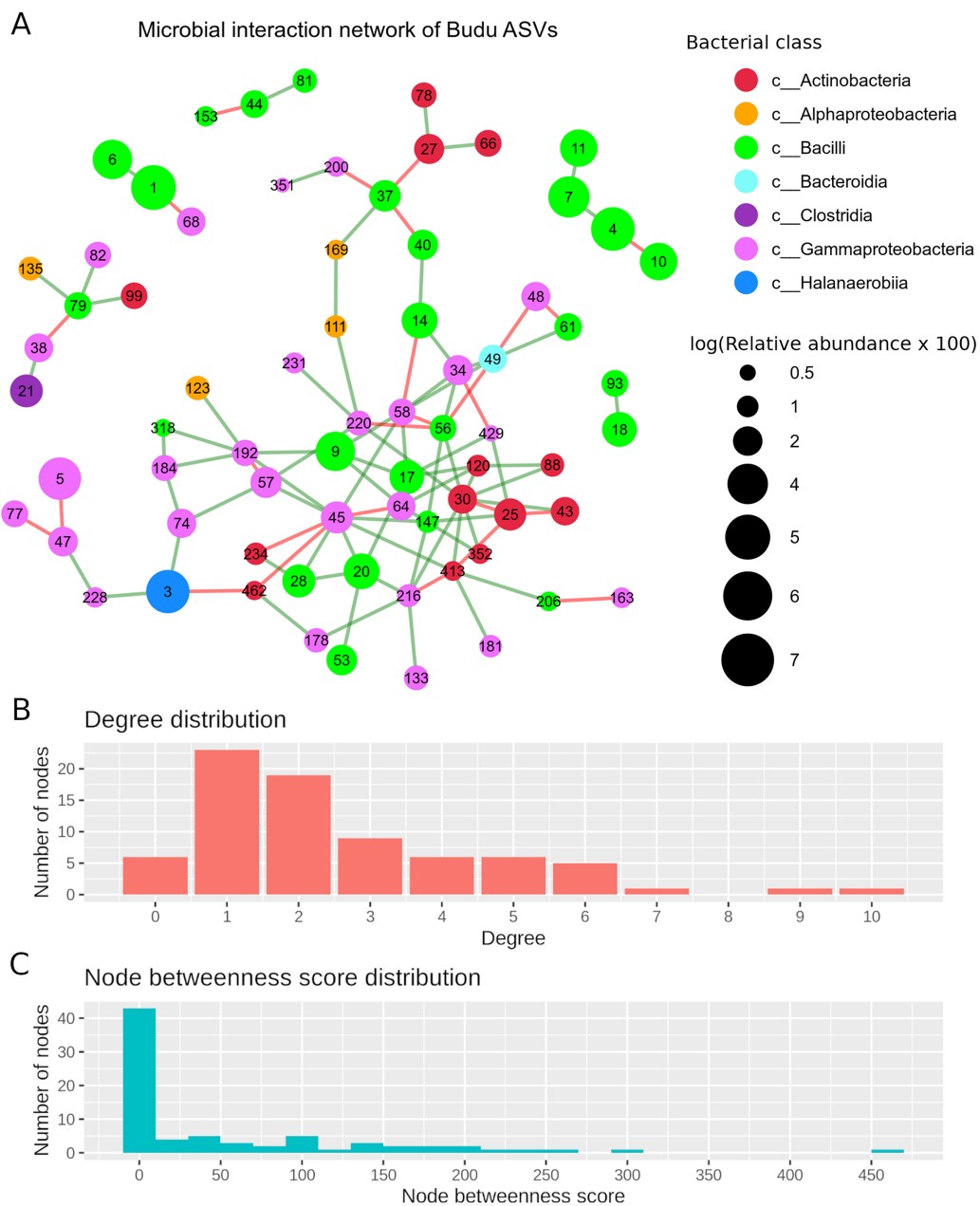

**Figure 5 Predicted microbial interaction network based on Meinshausen-Bühlmann neighborhood selection model applied on 16S sequence coabundance.** (A) Nodes represent ASVs, colored based on class taxonomic assignment, and node size was given based on relative abundance, scaled logarithmically. Green edges indicate positive interaction, while red edges indicate negative. To shorten the node label, the letters "ASV" was removed from the node label, (B) distribution of degree distribution, (C) distribution of degree betweenness scores.

ASV45 (Enterobacteriaceae) has the highest degree centrality (degrees = 10) and betweenness centrality (score = 466.72). It formed multiple links with other Gammaproteobacteria as well as Bacilli and Actinobacteria. *Tetragenococcus* nodes have relatively low node degrees and betweenness. The link formed between ASV1

*T. muriaticus* and ASV68 *Pseudomonas* was the only link predicted between the *Tetragenococcus* and non-*Tetragenococcus* nodes.

## DISCUSSION

Fermented fish sauce is a popular food condiment in the Southeast Asian region. The fermentation of raw fish is largely driven by its microbiota because the high salt concentration in fish sauce inhibits the activity of endogenous fish enzymes like cathepsins but activates halophilic bacterial proteinases (*Tungkawachara, Park & Choi, 2003*). There were also several microbiome-based research on fermented fish sauce in Korea that revealed further insights into the role of the microbiota in industrial fish sauce fermentation as well as fish sauce quality (*Jung et al., 2016*; *Lee, Jung & Jeon, 2014*). However, the current microbial understanding of the Malaysian fish sauce, *Budu* so far are based on culture-based methods, which only provide narrow taxonomic breadth and functional interpretation. One of the concerns that *Budu* research can address is that *Budu* consumption presents a potential health risk (*Mohd et al., 2011*; *Rosma et al., 2009*). Due to the rich purine content in anchovies (2,258.91 mg/kg) (*Li et al., 2019*), there is a concern that consumption of *Budu* could present a risk to higher gout incidence in Malaysia (*Paul & James, 2017*). Furthermore, based on *Rosma et al. (2009)*, seven of 12 *Budu* samples from 12 different producers contained histamine content exceeding 50 mg/100 g sample, which is the FDA limit for histamine consumption (*FDA, 2011*). Unchecked consumption of histamine-rich food can lead to scombroid food poisoning and the symptoms can include breathing difficulties and irregular heartbeat (*Tortorella et al., 2014*; *Wilson, Musto & Ghali, 2012*). By determining the microbial diversity and composition of *Budu*, we hope to start elucidating the relationship between the microbiota and *Budu* fermentation quality, which can later translate to enhancing the organoleptic and nutritional value of *Budu*.

There are apparent inconsistencies of *Budu* microbiota within brands in terms of phyla composition reiterated by the large distances in PCoA as well as differences in alpha diversity indices. This suggests that most *Budu* were produced under uncontrolled conditions between production batches which may lead to inconsistencies in the microbiota (*Jung et al., 2016*), and would be a major obstacle to commercial production with uniform, premium quality. From a food production point of view, inconsistent microbial diversity might interpret as inconsistent fermentation quality. Therefore, future research should include the microbiota as a fundamental component to assess consistency in fish sauce fermentation. However, these findings also mean a wider taxonomic diversity of bacteria can thrive in *Budu* and potentially contribute to *Budu* fermentation although at different quality.

*Tetragenococcus* is a tetrad-forming coccus, gram-positive halophilic lactic acid bacteria. It was previously isolated from high-salt fermented foods such as soy sauce, squid liver sauce (*Satomi et al., 1997*), dried fish, salted seafood (*Kim & Park, 2014*), fish sauce (*Thongsanit et al., 2002*) and fermented fish paste (*Marui et al., 2015*). Despite being the genus with the highest average relative abundance and prevalence, previous studies did not report its isolation from *Budu* (*Mohd Isa et al., 2020*; *Sim, Chye & Ann, 2009*).

Unsuccessful isolation of *Tetragenococcus* from *Budu* might be attributed to limited sampling or culturing conditions unsuitable to *Tetragenococcus*, showing the caveats of culturing-based methods. Despite its prevalence in numerous fermented food, to the best of our knowledge, *Tetragenococcus* was never found in the natural environment, in fish gut or in saltern sources (*Egerton et al., 2018*). Since Tetragenococcus possesses one copy of 16S rRNA gene per genome, the multiple *T. halophilus* and *T. muriaticus* ASVs found in our samples suggest the presence of different strains of the two species. The two species were reported to grow optimally at different pH levels and salt concentrations and utilize various types of sugar (*He et al., 2016*; *Kobayashi et al., 2004*; *Kobayashi, Kimura & Fujii, 2000*; *Udomsil et al., 2010*). *Tetragenococcus muriaticus* is known to release histamine and cadaverine, while *Tetragenococcus halophilus* is known to reduce histamine and cadaverine formation (*Kim et al., 2019*; *Zaman et al., 2010*). *T. halophilus* was also reported to contribute towards probiotic properties (*Kuda et al., 2014*). It is not yet understood how *Tetragenococcus* dominated a microbiome, or if it is possible to control its abundance in fermented food. Understanding the factors enabling *Tetragenococcus* dominance might translate to future applications including manipulation of the fermented food microbiome to enrich desirable metabolites or microbial processes.

The *Budu* microbiota composition was generally consistent. However, it was not stagnant, as shown by the significant increase of *Chromobacterium* in month 7, in samples from brand E and D. *Budu* from these two brands do not share any exclusive common ingredients. The two brands were also different in terms of microbiota composition, and therefore, we are unsure what factors caused the emergence of *Chromobacterium* in the two brands. Since *Chromobacterium* was not detected in the first and second batches, it is possible that the genus was introduced from the surrounding into the samples during sampling in months 0 and 3. *Chromobacterium* was previously found in environmental samples (*Kämpfer, Busse & Scholz, 2009*; *Menezes et al., 2015*) as well as food samples such as vegetables, cheese and seafood (*Koburger & May, 1982*). Bacteremia-causing *Chromobacterium* species were also reported (*Kaufman, Ceraso & Schugurensky, 1986*; *Parajuli et al., 2016*). Due to the lack of genomic sequences and analyses of the *Chromobacterium* species (*Santos et al., 2018*), future research should address this knowledge gap and identify the ecological significance of this genus in fermented food.

The microbiota determines the metabolism that takes place in *Budu*. By predicting the metagenome functions, *Tetragenococcus*-dominated samples are functionally different from non-*Tetragenococcus*-dominated samples. Non-*Tetragenococcus*-dominated samples were predicted to be enriched with the biosynthesis of amino acids and purine, which were known to contribute towards the quality of fermented foods in terms of pleasant organoleptic characteristics. Amino acids, such as glutamate and aspartate, and purines, such as inosinic acid and guanylic acid, contribute towards umami flavor (*Zhao et al., 2019*). These pathways are also implicated with health conditions. For instance, metabolites enriched in *Budu*, such as histamine and purine compounds, are implicated with health conditions such as scombroid poisoning and gout, respectively (*Paul & James, 2017*; *Tortorella et al., 2014*). Our results so far are predictive in nature; more omics data

such as transcriptomics metabolomics data are required to confirm the role of the microbiota on amino acid and purine biosynthesis in Budu.

A recent study demonstrated that amino acids and biogenic amines increased in fermented fish sauce after a year post-fermentation and was also influenced by storage conditions such as temperature (*Joung & Min, 2018*). Another study showed that prolonged fermentation of fish could also lead to the conversion of glutamic acid in the fish protein to more purines (*Tungkawachara, Park & Choi, 2003*). However, we did not observe changes in predicted enriched pathways across sampling batches.

Microbes exist in communities and interact through mechanisms such as production of lactic acid, releasing of antimicrobials, quorum sensing, and cross-feeding. The ability to broadly predict such relationships using relative abundance information from 16S sequencing is alluring, thus attempts on modeling microbial interactions were done in the form of microbial interaction network (*Deng et al., 2012*; *Faust et al., 2012*; *Friedman & Alm, 2012*; *Kurtz et al., 2015*). For instance, the is only one link that formed between *Tetragenococcus* nodes and non-*Tetragenococcus* nodes. This prediction could mean one of two things. Since the network relies on relative abundance information, the interaction network prediction could not model in situations where ASVs are highly dominant. On the other hand, it could mean that *Tetragenococcus* genuinely form only a few interactions with other taxa. Unfortunately, the only validated microbial interaction made by *Tetragenococcus* was an antagonistic interaction with *Zygosaccharomyces rouxii* which is a yeast (*Devanthi et al., 2018*). The interpretation of microbial interaction network and its application in a biological context is still limited due to strong environmental effects which obscures genuine microbial interactions (*Röttjers & Faust, 2018*). What is needed is a gold standard model microbial system to benchmark predicted interactions. *Budu* easily lends itself as a tractable microbial system that is replicable, manipulable and cheap, making it a potential gold standard for benchmarking current and future microbial interaction network models (*Wolfe, 2018*; *Wolfe & Dutton, 2015*).

## CONCLUSIONS

The microbiota of some brands of *Budu* was inconsistent, which suggests that *Budu* production was done under uncontrolled conditions between production batches. This study demonstrated the utility of 16S amplicon sequencing as a viable, cost-effective method for quality assurance of fermented food. We also discovered abundant and prevalent genera of *Budu*. Albeit not reported in prior culture-dependent studies on *Budu*, *Tetragenococcus* was the most abundant genus in our sample collection, although devoid in some samples. *Tetragenococcus* is well adapted to the fermentation environment, but a greater diversity of bacteria are as capable as *Tetragenococcus* in fulfilling its role in fermentation, opening up possibilities of exploring a wider array of microbes as a candidate starter culture. Non-*Tetragenococcus* dominated samples were predicted to be enriched with metabolic pathways associated with biosynthesis of amino acid biosynthesis and purine, metabolites that were attributed to organoleptic properties as well as health. We also detected *Chromobacterium* as the only genera that significantly increased in the last sampling batch, though its ecological role in Budu and its potential source is not

clear. We also attempted to model the microbial interaction using 16S abundance data, although its interpretation needs to be validated.

## ACKNOWLEDGEMENTS

We are grateful to Wan Nur Afiq and Wan Nur Fariees Fitrie for providing Budu samples. We want to thank Monash University Malaysia Genomics Facility for providing computational resources. We are also thankful to Amazon Web Services ASEAN Public Sector, particularly Sammy Lock and Shazli Mohd Ghazali for provision of proof of concept (POC) credits which we used for computationally intensive microbial interaction prediction. We are fortunate to have Yin Peng Lee (ORCID ID: 0000-0002-3395-1385) in assisting with the wet lab operations. Finally, we express our gratitude to Fadilla Wahyudi (ORCID ID: 0000-0002-0454-6049), Shu Yong Lim (ORCID ID: 0000-0003-2232-8261), Sze Mei Lee (ORCID ID: 0000-0002-9089-6490) and Yit Kheng Goh (ORCID ID: 0000-0003-2884-7273) for reviewing the manuscript and suggesting changes.

### Funding

Monash University Malaysia Genomics Facility provided financial support for this project. The funders had no role in study design, data collection and analysis, decision to publish, or preparation of the manuscript.

### Grant Disclosures

The following grant information was disclosed by the authors:
Monash University Malaysia Genomics Facility.

### Competing Interests

Han Ming Gan is employed by GeneSEQ Sdn Bhd.

### Author Contributions

- Muhammad Zarul Hanifah Md Zoqratt conceived and designed the experiments, performed the experiments, analyzed the data, prepared figures and/or tables, authored or reviewed drafts of the paper, and approved the final draft.
- Han Ming Gan conceived and designed the experiments, authored or reviewed drafts of the paper, and approved the final draft.

### DNA Deposition

The following information was supplied regarding the deposition of DNA sequences:
All FastQ raw data is available at SRA (SRP193353) and at NCBI BioProject (PRJNA534025).

### Data Availability

The raw data is available in the Supplemental Files.

## Supplemental Information

Supplemental information for this article can be found online at http://dx.doi.org/10.7717/peerj.12345#supplemental-information.

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
