# Peer review of "The inconsistent microbiota of Budu, the Malaysian fermented anchovy sauce, revealed through 16S amplicon sequencing"

_PeerJ, doi:10.7717/peerj.12345_

## Round 0.1 · original submission · Minor Revisions

The reviewers are largely positive. There are a couple of editorial comments that the authors may wish to take inito account.

Reviewer 1 ·

Basic reporting

Article is logically consistent and clearly written.

Experimental design

Methods are well described, most of samples have biological replicates. The only drawback is the lack of negative controls that can help to catch possible contaminations (for example potential contamination with Chromobacterium during sampling). However microbiome composition is very specific and do not look as suffered from contamination.

Validity of the findings

no comment

·

Basic reporting

A manuscript describes an analysis of the microbiota of a Budu, a Malasyan sauce produced by fermentation of anchovies. Because fermentation, in this case, is a microbial-based process, the structure of the microbiota may be crucial for the features of the final product, including its impact on consumer health.
The manuscript is quite interesting and devoted to a rather unusual topic, as like food-producing microbiota. The work is well done and the stated goals are achieved successfully.
The text of the manuscript is written clearly and it was interesting for me to read it.
Finally, I recommend accepting this manuscript after some minor revisions.

Experimental design

The experimental design is correct and well thought out. The workflow includes sample collection (60 samples from 20 bottles of the sauce), amplification and analysis of the 16S rRNAs, the definition of the taxonomical structure of the samples, description of the positive and negative correlations for different species of microbes, and prediction of the overrepresented pathways.

Validity of the findings

The results of this research can be interesting for microbiologists and biochemists, as well as for specialists in the food industry.

Additional comments

Minor comment:
Lines 59-60: "The fish sauce microbiota can change the fish sauce metabolite content, potentially altering its health and gastronomic properties." - Absolutely unclear what are there "health properties" of the sauce. Please change this sentence to make it clear.

Desires:
1. I would like to see a more detailed explanation for why is it important to study the microbiota of the Budu sauce. I mean, why this sauce is so important to study its microbiota?
2. I would like to see a more detailed description of functional and pathway metagenome prediction. What pathways were also enriched, in addition to the listed ones? How different are the analyzed samples on the level of enriched pathways? Is a taxonomic variability of samples correlate with a pathway variability?

---

## Round 0.2 · accepted · Accept

The reviewers' comments have been addressed.